# Differences in BNT126b2 and ChAdOx1 Homologous Vaccination Antibody Response among Teachers in Poznan, Poland

**DOI:** 10.3390/vaccines11010118

**Published:** 2023-01-03

**Authors:** Dagny Lorent, Rafał Nowak, Monika Jankowska, Łukasz Kuszel, Paweł Zmora

**Affiliations:** 1Institute of Bioorganic Chemistry Polish Academy of Sciences, 61-704 Poznan, Poland; 2Department of Medical Genetics, Poznan University of Medical Sciences, 60-806 Poznan, Poland

**Keywords:** SARS-CoV-2, antibodies, teachers, seroprevalence, vaccine, Poland

## Abstract

Children are among the best vectors to spread respiratory viruses, including emerging variants of SARS-CoV-2 due to the asymptomatic or relatively mild course of infection and simultaneously high titres of pathogens in the respiratory tract. Therefore, individuals who have constant contact with children, e.g., teachers should be vaccinated against COVID-19 as essential workers within the first phases of a vaccination campaign. In Poland, primary and secondary school teachers were vaccinated with ChAdOx1 from February 2021 with a three month interval between the two doses, while lecturers at medical universities, who are simultaneously healthcare workers, received the BNT126b2 vaccine from December 2020 with three weeks between the first and second doses. The aim of this study was to compare the antibody responses at two weeks and three months after vaccination and to estimate the vaccine effectiveness against COVID-19 among infection-naïve teachers vaccinated with mRNA and a vector vaccine. We found that the anti-SARS-CoV-2 spike protein antibodies were significantly higher among the lecturers but antibody waning was slower among the schoolteachers. However, those vaccinated with ChAdOx1 complained significantly more often of vaccine side effects. In addition, during the three months after the second vaccine dose no study participants were infected with SARS-CoV-2. The BNT126b2 vaccine gave higher antibody titres in comparison with ChAdOx1 but protection against COVID-19 in both cases was similar. Moreover, we did not find any anti-SARS-CoV-2 nucleoprotein antibodies at two weeks as well as at three months after vaccination among the study participants, which shows a very high vaccine effectiveness in the occupational group with a high SARS-CoV-2-infection risk.

## 1. Introduction

There is an urgent need to address the health problems associated with the coronavirus disease 2019 (COVID-19) pandemic caused by severe acute respiratory syndrome coronavirus 2 (SARS-CoV-2). Since its outbreak in December 2019 in Wuhan, China, the WHO has reported over 630 million confirmed COVID-19 cases and 6.58 million COVID-19 related deaths worldwide with almost 6.3 million cases and approximately 118 thousand deaths in Poland (as of November 2022) [1]. Global efforts to prevent SARS-CoV-2 transmission have led to the development of effective vaccines based on different platforms: mRNA in liposomes, viral vectors, and inactivated viruses [2,3,4]. As of 22 November 2022, there have been 172 vaccine candidates in clinical development [5] and the European Medicines Agency has authorised six vaccines for use, i.e., BNT162b2 (Pfizer, New York, NY, USA/BioNTech, Mainz, Germany), mRNA-1273 (Moderna, Cambridge, MA, USA), ChAdOx1 (Astra Zeneca, Cambridge, UK/Oxford University, Oxford, UK), Ad26.COV2.S (Janssen Pharmaceutical Companies, Beerse, Belgium), NVX-CoV2373 (Novavax, Gaithersburg, MD, USA), and VLA2001 (Valneva, Vienna, Austria) [6]. However, at the beginning of the vaccination roll-out in December 2020 the types of vaccines and number of doses available were limited. Therefore, many countries scheduled priority groups based on SARS-CoV-2 infection and COVID-19 severity risks [7,8].

The Polish national vaccination strategy was divided into four phases. Initially, vaccines were administered to healthcare workers (HCWs), social care workers, and medical students (phase 0); then vaccines were offered to individuals over 60 years of age, residents in long-term care facilities, and public service workers (phase I); next to adults with comorbidities and other essential workers (phase II); and finally to persons over 16 years of age (phase III) [9]. During the vaccination campaign in Poland, five vaccine products were deployed: BNT162b2 since 23 December 2020; ChAdOx1 and mRNA-1273 since 6 January 2021; Ad26.COV2.S since 3 February 2021; and NVX-CoV2373 since 2 March 2022. Different COVID-19 vaccine products were recommended for specific priority groups and adjusted according to the current epidemiological situation [9]. The BNT162b2 vaccines were administrated to HCWs since 27 December 2020, whereas teachers were receiving ChAdOx1 vaccines since 12 February 2021 [9]. Both BNT162b2 and ChAdOx1 elicit immune responses mainly against the receptor-binding domain (RBD) of the spike (S) protein. However, the S protein consists, additionally of the following domains: N-terminal domain (NTD), fusion peptide (FP), two heptad repeats (HR1 and HR2), a transmembrane domain (TM), and a cytoplasmic tail (CT), which may be also a target for antibodies produced as a result of vaccination. The structure of the SARS-CoV-2 S protein is well presented by Huang et al. [10]. It should also be highlighted that different vaccine types utilize different nucleotide sequences as well as delivery methods. ChAdOx1 encodes a full-length S protein identical to the SARS-CoV-2 wild*-*type (wt) strain, i.e., Wuhan-Hu-1 [11]. In contrast, the BNT126b2 contains nucleoside modified mRNA with pseudouridines and differs in two amino acids from the wt S protein sequence; the different amino acids stabilize the construct in the cellular environment and stabilize the prefusion conformation of the S protein [12]. The ChAdOx1 is delivered in the replication-deficient chimpanzee adenovirus-vector, whereas BNT162b2 contains the nucleoside modified mRNA sequence encapsulated in lipid nanoparticles [11,12]. Both vaccines are administrated intramuscularly and were shown to be effective in the delivery of viral antigens, thereby stimulating the host immune system to recognize SARS-CoV-2. Nevertheless, the response of the immune system to COVID-19 vaccination as well as the duration of protection is not fully understood. Therefore, the aim of this study was to determine the anti-SARS-CoV-2 IgG antibody levels in infection-naïve education workers in primary schools and in medical universities after homologous ChAdOx1 or BNT162b2 vaccination, and to estimate the vaccines effectiveness against COVID-19 in those occupational groups.

## 2. Materials and Methods

### 2.1. Study Participants and Design

We invited approximately 100 teachers from several primary schools in Poznan, as well as 100 academic teachers from Poznan University of Medical Sciences (PUMS) to participate in our study and obtained a positive response from 36 and 45 individuals, respectively. The primary school teachers were vaccinated twice with ChAdOx1 vaccine with a three month interval between the two doses. The PUMS academic teachers, who simultaneously were HCWs (physicians and laboratory diagnosticians), were vaccinated with the two doses of BNT162b2 mRNA COVID-19 vaccine with a three week interval between the first and second dose. 

Written questionnaires were used to collect data on sex, age, comorbidities, COVID-19 history, and the administered vaccine type from all the participants. Additionally, the study participants were asked about the side effects following the vaccination.

Blood specimens were collected from all the enrolled study participants at the IBCH PAS, Poznan, Poland, two weeks and approximately three months after the completion of the homologous vaccination schedule. Unfortunately, 19 out of the 35 primary school teachers and 16 out of the 45 PUMS academic teachers did not attend, did not answer our contact attempts, and thus did not donate their blood at the second time point.

### 2.2. Laboratory Analysis

The anti-SARS-CoV-2 IgG antibody levels after vaccination were measured using quantitative SARS-CoV-2 IgG QuantiSpike ELISA assay (TK040, Vitrotest Europe, Wroclaw, Poland) targeting spike (S) protein. To test whether the study participants were infected with SARS-CoV-2 we used rapid cassette tests: 2019-novel coronavirus IgG/IgM antibody detection kit (Vazyme), which detects anti-SARS-CoV-2 N antibodies generated only after natural infection. Tests were performed and interpreted according to the manufacturer’s instructions.

### 2.3. Statistical Analysis

All statistical analyses were performed using GraphPad Prism 9 software. The differences between the two groups were analysed by the Mann–Whitney test. The data were accepted as statistically different if *p* < 0.05.

### 2.4. Ethics Approval

The study was approved by the Bioethics Committee at the Poznan University of Medical Sciences, Poznan, Poland (Resolution No. 470/20 from June 2019). In addition, written informed consent was obtained from each of the study participant before blood collection.

## 3. Results

### 3.1. Characteristics of the Study Participants

The study group consisted of 81 teachers: 36 from schools and 45 from PUMS (Table 1). Most of the participants were female: 91.67% (*n* = 33) from schools and 86.67% (*n* = 39) from PUMS. From the schools, the mean age of the enrolled individuals was 45.3 ± 11.7 years and over half (52.78%, *n* = 19) did not report any comorbidities, whereas 33.3% (*n* = 12) suffered from circulatory system chronic diseases (CDs), 11.11% (*n* = 4) from autoimmunological CDs, 5.56% (*n* = 2) respiratory system CDs, and 2.78% (*n* = 1) had a mental disorder. The PUMS lecturers were aged 37.9 ± 13.2 years on average and most (80%; *n* = 36) were healthy; however, some individuals had circulatory (8.89%, *n* = 4), respiratory (8.89%, *n* = 4), autoimmunological (2.22%, *n* = 1) CDs, and 2.22% (*n* = 1) had a mental disorder (Table 1). Notably, none of the participants reported a history of COVID-19 and did not produce anti-SARS-CoV-2 N antibodies as a result of natural infection (Table 1).

### 3.2. Anti-SARS-CoV-2 IgG S Antibody Levels at Two Weeks after the Second Dose of the COVID-19 Vaccine

Regardless of the vaccine type, all the study participants developed anti-SARS-CoV-2 S IgG antibodies two weeks after the second vaccine dose. The antibody levels in the BNT162b2 vaccinated individuals ranged from 388 to 3910 BAU/mL, and in the ChAdOx1 recipients from 33 to 1897 BAU/mL (Figure 1). The average anti-SARS-CoV-2 S IgG antibody levels were significantly higher among the BNT162b2 vaccinated individuals (1492 ± 700 BAU/mL) in comparison with the ChAdOx1 recipient group (426.9 ± 461 BAU/mL) (Figure 1).

### 3.3. Side Effects after Two Doses of Either BNT162b2 or ChAdOx1

After the two BNT162b2 vaccine doses, 6.67% (*n* = 3) of recipients reported at least one side effect. The most commonly reported side effects were fever (6.67%, *n* = 3), followed by chills (2.22%, *n* = 1), fatigue (2.22%, *n* = 1), and headache (2.22%, *n* = 1). In contrast, in the ChAdOx1 group 72.22% (*n* = 26) of the recipients reported at least one side effect after two vaccine doses. More than half had fever (52.78%, *n* = 19), followed by muscle and joint pain (41.67%, *n* = 15), headache (36.11%, *n* = 13), fatigue (19.44, *n* = 7), chills (16.67, *n* = 6), pain at the injection site (11.11%, *n* = 4), diarrhoea (2.78%, *n* = 1), and vomiting (2.78%, *n* = 1). Thus, for both vaccine types, fever was the most frequent side effect. However, when compared with the BNT162b2 group, side effects were more frequent and varied in the ChAdOx1 group (Figure 2).

### 3.4. Anti-SARS-CoV-2 IgG S Antibody Levels Three Months after Vaccination

In the follow-up study, 29 of the 45 participants vaccinated with BNT162b2 and 16 of the 35 vaccinated with ChAdOx1 provided blood samples. Three months after the second vaccine dose administration, eight study participants vaccinated with BNT162b2 and one individual from the ChAdOx1 vaccinated group did not have any detectable anti-SARS-CoV-2 S IgG antibodies. In addition, we observed that the participants who remained seropositive exhibited a significant decline in antibody titres (Figure 3a). Within 10–12 weeks after the second vaccine dose, the antibody levels in the BNT162b2 recipients decreased by 90.8 ± 11.6% and by 67.28 ± 21.98% in the ChAdOx1 recipients (Figure 3b). Three months after vaccination, none of the analysed study participants developed anti-SARS-CoV-2 N antibodies, which shows the very high effectiveness in protection against COVID-19 of both vaccines.

## 4. Discussion

To our knowledge this is the first study of the anti-SARS-CoV-2 IgG S antibody response following COVID-19 vaccination among education workers in Poland. Lecturers and primary and secondary school teachers were in the second vaccination priority group in Poland, except those academics who were simultaneously HCWs employed at a medical university who were vaccinated during the so-called phase 0 [9]. According to the UNESCO/UNICEF/World Bank/OECD Survey on National Education Responses to COVID-19 School Closure, 72% of countries (146 of 204) prioritised teachers for vaccination; however, the vaccination rate differs widely among countries with the highest proportions of vaccinated teachers being in high-income countries, e.g., Portugal, Chile, Sweden, and Saudi Arabia, while relatively low teacher vaccination rates were found in low-income countries, e.g., Algeria and Venezuela [4]. According to the Polish Ministry of Education and Science more than 90% of primary and secondary school teachers and 89% of lecturers were administered with the first dose of the vaccine against COVID-19 in February 2022, while the general vaccination rate in Poznan at the beginning of our study was estimated as 3.84%[9,13]. Education workers are among the most vaccinated occupational groups in Poland and the percentage of individuals fully protected against severe COVID-19 is much higher than that of the general Polish population (57.8%) [13]. Unfortunately, there are no detailed data available on primary and secondary school teacher vaccinations (e.g., differences in the number of vaccinated education workers between primary and secondary schools) or comparisons of vaccination rates in specific regions. However, taking into consideration that Poznan, and the Greater Poland region, has one of the highest levels of vaccinated citizens [13], we may assume that vaccination rates among teachers are also among the highest. The rates among the PUMS academics and the PhD students enrolled in teaching are in the top ten among the universities in Poland at 94.7% (1670/1763) and 97.3% (180/185), respectively [9].

According to the Polish COVID-19 vaccination strategy, education workers received two doses of ChAdOx1 during phase 1 (from February 2021). It consists of a non-replicating adenoviral vector encoding for the SARS-CoV-2 S protein, with a 4–12 week dosing interval [14]. The clinical trials indicated a 76% efficacy after administration of the first dose and 81% after the second dose, with the interval extended to ≥12 weeks. Dosing intervals shorter than six weeks resulted in a lower vaccine efficacy, i.e., 55% [15]. An exception was made for medical university lecturers who were simultaneously HCWs who were administered with the BNT162b2 mRNA vaccine in the so-called phase 0 (from December 2020) This vaccine contains nucleoside-modified mRNA encoding for the full-length SARS-CoV-2 S protein encapsulated into lipid nanoparticles, which should be administered in two doses within 21 days [16]. The clinical trials demonstrated a 52% efficacy in preventing COVID-19 after the first vaccine dose and 95% after the full vaccination [17], which lasted up to six months. In our study, both BNT162b2 and ChAdOx1 recipients developed detectable anti-SARS-CoV-2 S IgG antibodies two weeks after the second vaccine dose. However, antibody levels were higher after the BNT162b2 vaccination than after the ChAdOx1 vaccination. In contrast, three months after the two homologous vaccine doses the antibody levels waned more rapidly in the BNT162b2 recipients when compared to the ChAdOx1 vaccinated individuals. Similar trends were observed among the HCWs vaccinated with the BNT126b2 or ChAdOx1 vaccine in Cyprus [18]. Our results are also in line with a large-population–based analysis by Wei et al., who demonstrated that following the homologous BNT162b2 vaccination anti-SARS-CoV-2 S IgG antibodies titres were higher but waned faster over time in comparison to vector vaccination[19,20]. Higher anti-SARS-CoV-2 S antibody titres among the BNT126b2 vaccinated individuals in comparison to the ChAdOx1 recipients were also observed by Barin et al., but in contrast to our results, they found the lowest decline of antibody levels among the mRNA vaccine recipients [21]. What remains unexplained is that in some participants, the post-vaccination antibodies were not detectable three months after the second vaccine dose even though they were present two weeks post-vaccination. The absence of antibodies was more common in the BNT162b2 than in the ChAdOx1 recipients. The BNT162b2 vaccinated group primarily included young and healthy persons, whereas the ChAdOx1 recipients were older and had a greater number of comorbidities. Thus, our results significantly differ from earlier findings, where lower antibody levels after vaccination were associated with comorbidities, male sex, and older age [20,22]. Wei et al. reported that the above-mentioned factors had larger effects on the BNT162b2 response than on the ChAdOx1 response but did not affect the half-life regardless of the vaccine type. The estimated half-life of post-vaccination antibodies was 79 days for ChAdOx1 and 51 days for BNT162b2 [20]. Nevertheless, at present, the cause of the rapid drop in anti-SARS-CoV-2 S IgG antibodies in some participants remains unclear.

In addition to studies focusing on the antibody response after vaccination, several cohort studies were conducted to compare the effectiveness of the ChAdOx1 and BNT162b2 vaccines. Kaura et al. revealed that the SARS-CoV-2 infection incidence rate ratio after a single vaccine dose was higher for ChAdOx1 than for BNT162b2, although the vaccines were equally effective at reducing COVID-19 mortality and hospitalisation rates [23]. Similar observations were made for people who completed the full vaccination, with two doses [23]. At 14–73 days post-vaccination, Hall et al. observed lower effectiveness for ChAdOx1 when compared to the BNT162b2 vaccine, i.e., 58% and 85%, respectively [24]. Two studies conducted five months after the second-dose administration reported higher effectiveness for BNT162b2 than for ChAdOx1 [25,26]. However, Andrews et al. noted that protection against COVID-19-related death and hospitalisation remained high, regardless of the vaccine type [26]. Several groups have found that the rate of decline in vaccine effectiveness was higher in older individuals and those with comorbidities [25,26,27]. Tartof et al. suggested that reduction in vaccine effectiveness may be a result of waning immunity with time [28].

Due to the above-described depletion of anti-SARS-CoV-2 S antibodies and thus potential weaker protection from severe COVID-19, as well as the emergence of the SARS-CoV-2 omicron variant, a third vaccine dose, the so-called ‘booster’, is recommended by the EMA, the European Center for Disease Prevention, and many other scientific and medical societies. A third vaccine dose was reported to elevate the anti-SARS-CoV-2 S antibody level, increasing the neutralising antibodies titre, and showing activity against all the circulating SARS-CoV-2 variants, thus reducing the risk of infection by 88% to 92% [29,30,31,32,33]. Unfortunately, protection against severe COVID-19 after three vaccine doses wanes in a similar manner to the previous dose, especially among older people with comorbidities and patients on immunosuppressants [34,35,36]. Therefore, in the case of individuals who are at risk of progression to severe COVID-19, i.e., adults over 60 years of age, immunocompromised individuals, and pregnant women both the EMA and ECDC recommend a fourth vaccine dose (second booster) [37].

Overall, in our study, we did not observe vaccine-specific differences in protection against COVID-19 after two vaccine doses, as both vaccines provided high immunity to SARS-CoV-2 in infection-naïve education workers. No teachers or lecturers included in our study manifested COVID-19 within three months after the full vaccination. Moreover, those vaccinated with ChAdOx1 and included in our study were exposed to unvaccinated and potentially SARS-CoV-2-infected pupils, since the BNT126b2 vaccines were approved for 12–15-year-old children in June 2021, 6–12-year-old children in December 2021, and children below six years of age in November 2022. The low incidence of SARS-CoV-2 infection among those vaccinated with ChAdOx1 may be a result of vaccination as well as the implementation of various prevention strategies, including complete school closures. The duration of school closures depended on the current epidemiological situation and varied by school type. In general, the higher the educational stage, the longer the period of school closure. Primary schools were closed from 9 November 2020 to 17 January 2021 and between 11 March and 30 April 2021 with children having only online classes. In addition, in primary schools when one child became infected with SARS-CoV-2, the entire group was quarantined. However, it should be underlined that there are still considerable uncertainties regarding the role of children in SARS-CoV-2 transmission. A meta-analysis by Caini et al. suggested a limited viral spread in school settings. In addition, the authors postulated that children were less than half as likely to have been tested seropositive in comparison to adults [38]. The lower susceptibility of children to SARS-CoV-2 transmission relative to adults was also demonstrated in a meta-analysis conducted by Viner et al. [39]. Moreover, several studies have shown that seroprevalence in educational settings reflects the local SARS-CoV-2 infection rate and is probably associated with general community transmission [40,41,42,43,44]. In contrast to primary and secondary schools, teaching at the university level moved almost completely to the remote mode during the first waves of the COVID-19 pandemic, and thus, they should represent a lower SARS-CoV-2 infection risk. However, due to their medical background, the lecturers involved in our study were also involved in fighting the pandemic, e.g., in molecular diagnostics of SARS-CoV-2 infection. Our previous study showed that HCWs are at the highest risk of SARS-CoV-2 infection and are characterised by the highest seroprevalence among the occupational groups [45]. Further, other authors have demonstrated the clinical work environment as a key SARS-CoV-2 infection risk factor [46,47] but have also highlighted that local COVID-19 foci and household contacts may affect seropositivity rates [47,48]. Surprisingly, none of the analysed lecturers were infected with SARS-CoV-2 during the three months after full vaccination, which can be explained by the high efficiency of the BNT126b2 vaccine, as well as good practical preparation, knowledge, experience, and the proper equipment to work with samples from patients potentially infected with SARS-CoV-2.

Any drug may have side effects, and the COVID-19 vaccines are not an exception. Therefore, we analysed the severity of the side effects following the second ChAdOx1 or BNT162b2 vaccine dose. As stated in the BNT162b2 product information, following the second dose, the most frequent adverse reactions were injection site pain, fatigue, and headaches. Less than half of the BNT162b2 recipients experienced myalgia, chills, arthralgia, pyrexia, or injection site swelling. The increased risk of myocarditis in younger males has also been highlighted [16]. In our study, after two BNT162b vaccine doses, adverse effects occurred at much lower frequencies than expected from the published literature and manufacturer’s data. At least one side effect, such as fever, headache, fatigue, or chills was observed in 7% of the BNT162b2 recipients. An observational study conducted by Menni et al. in 2021 demonstrated that the incidence of systemic side effects after the second BNT162b2 vaccine dose was lower than that for local side effects (22% versus 69%, respectively), but both were less prevalent than suggested in the product information A higher local reactogenicity was associated with previous SARS-CoV-2 infection. The side effects were most commonly reported by younger women [49]. Coggins et al. noted a similar trend regarding age and gender in generally healthy and SARS-CoV-2 infection-naïve HCWs. The authors also demonstrated that post-vaccination symptoms were negatively correlated with body weight. In our study, the BNT126b2 vaccinated group consisted mainly of young and healthy lecturers from medical universities who may tend to underestimate symptom intensity. In contrast, the ChAdOx1 vaccine raised more safety concerns. In March 2021, due to thromboembolic events, several countries paused vaccination with ChAdOx1. Many resumed the use of vaccines following the EMA’s safety committee statement that the benefit of ChAdOx1 vaccination in protecting against COVID-19 outweighs the risks [50]. According to ChAdOx1 product information, the most commonly reported adverse reactions are injection site tenderness and pain, headaches, and fatigue. In less than half of the cases, myalgia, malaise, pyrexia, chills, arthralgia, and nausea are observed. Thrombosis with thrombocytopenia syndrome was reported as a very rare case [14]. In our research, in contrast to the BNT162b2 group, systemic reactogenicity was more common and severe after two doses of ChAdOx1. The majority (77%) of the ChAdOx1 recipients experienced multiple adverse reactions, such as fever, muscle and joint pain, headache, fatigue, chills, pain at the injection site, diarrhoea, or vomiting. Warkentin et al. reported at least a reaction in 58% of ChAdOx1 recipients, 70% of whom had pre-existing diseases. In addition, higher reactogenicity was observed among women and younger people; in contrast, 72% of the ChAdOx1 recipients in a smaller study conducted by Marking et al. did not report any adverse effect [51]. Teachers vaccinated with ChAdOx1 are older and many suffer from underlying diseases, so they may pay greater attention to their health after vaccination. Notably, as side-effects studies rely mostly on self-reported data, we may assume that this could lead to information bias among different studies and population groups.

## 5. Conclusions

In conclusion, the strengths of our study include the comparison of antibody responses to ChAdOx1 and BNT162b2 homologous vaccination in terms of effectiveness, safety, and antibody levels at two time points. We also evaluated both the prevention and control measures implemented in educational settings, which altogether, allowed us to estimate immunity in two groups of teachers. We are aware that our study has limitations. It includes a relatively small sample size as well as age and gender disparities. On the other hand, the majority of school teachers in Poland are women (88%), of which 62% are over 40 years old [52]. In fact, our study reflects the demographics of education workers, thus providing an insight into the COVID-19 immunity status of this group. Another limitation is the relatively short follow-up. Further data collection would be needed to determine exactly how changes in IgG antibody levels over a longer period affect the protection against SARS-CoV-2 infection. It should also be highlighted that our research focuses on antibody responses. Studying other mechanisms, such as memory T-cells may enhance our understanding of vaccine-induced immunity. Lastly, due to the very high vaccination rate, we were unable to collect samples from the non-vaccinated teachers, which could serve as a control group in comparison to the vaccine effectiveness and SARS-CoV-2 infection risk.

## Figures and Tables

**Figure 1 vaccines-11-00118-f001:**
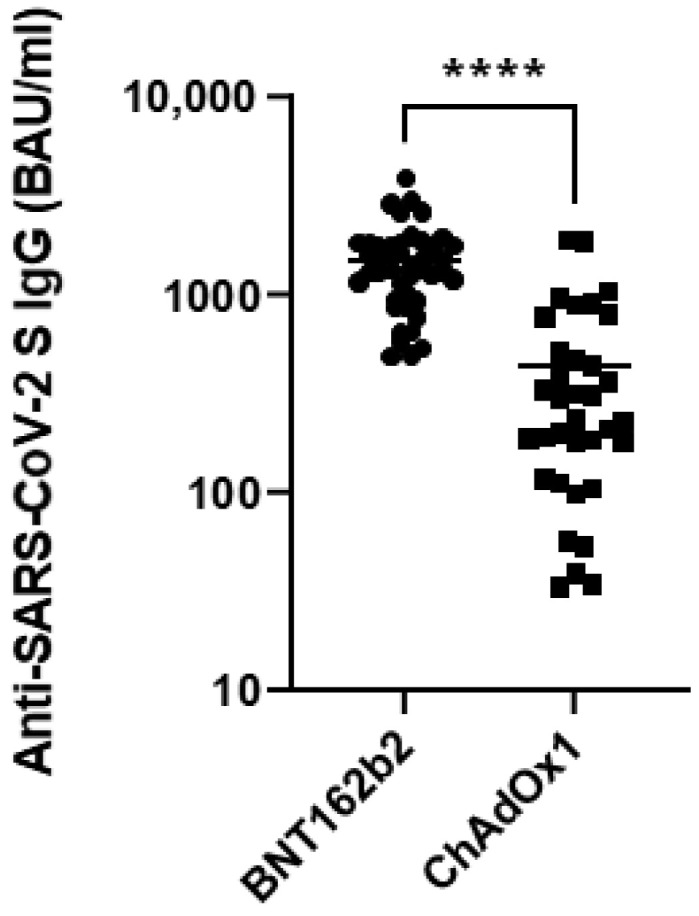
Levels of anti-SARS-CoV-2 spike protein (S) IgG antibodies two weeks after the two homologous doses of BNT162b2 or ChAdOx1 vaccines. ****—*p* < 0.001.

**Figure 2 vaccines-11-00118-f002:**
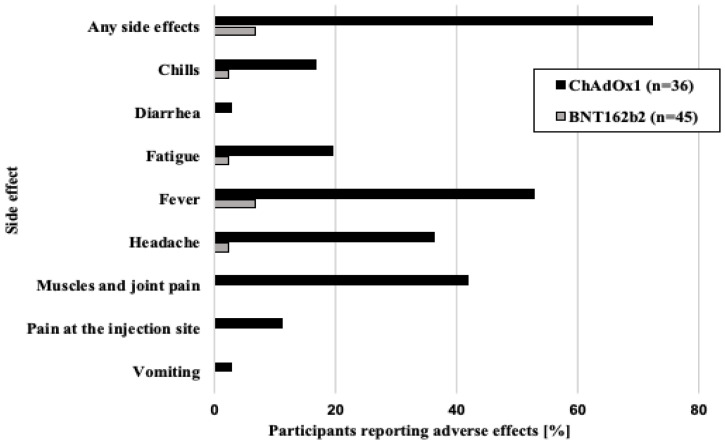
Side effects after the two doses of BNT62b2 or ChAdOx1 vaccine. Participants could report more than one side effect.

**Figure 3 vaccines-11-00118-f003:**
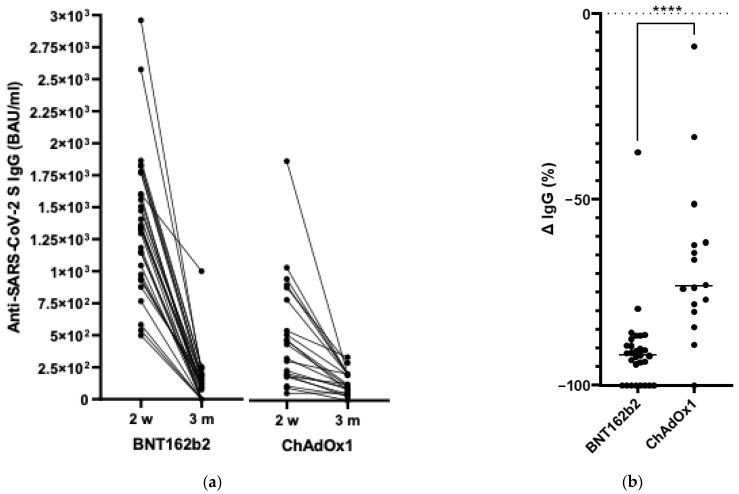
The anti-SARS-CoV-2 spike protein IgG antibody waning among teachers vaccinated with BNT162b2 or ChAdOx1 (**a**) from two weeks to three months after the second vaccination second dose, and (**b**) presented as a percentage change at three months post vaccination. ****—*p* < 0.001.

**Table 1 vaccines-11-00118-t001:** Characteristics of the study participants.

	Study Participants	
	Primary School Teachers	Academic Teachers
Number of participants	36 (100%)	45 (100%)
Sex		
Female	33 (91.67%)	39 (86.67%)
Male	3 (8.33%)	6 (13.33%)
Age		
<30 y.o.	6 (16.67%)	15 (33.33%)
31–40 y.o.	3 (8.83%)	12 (26.67%)
41–50 y.o	12 (33.33%)	12 (26.67%)
51–60 y.o.	13 (36.10%) ^a^	2 (4.44%) ^b^
>60 y.o.	2 (5.56%)	4 (8.89%)
Comorbidities and CDs		
None	19 (52.78%) ^a^	36 (80.00%) ^b^
Circulatory system CDs, i.e., hypertension	12 (33.33%)	4 (8.89%)
Respiratory system CDs, i.e., asthma	2 (5.56%)	1 (2.22%)
Infectious CDs, i.e., HIV/AIDS	-	-
Chronic kidney diseases	-	-
Digestive track CDs, i.e., Crohn diseases	-	-
Autoimmunological CDs, i.e., allergies	4 (11.11%)	4 (8.89%)
Neoplasmic diseases, i.e., cancer	-	-
Metabolic diseases, i.e., diabetes	-	-
Mental disorders, i.e., depression	1 (2.78%)	1 (2.22%)
COVID-19		
COVID-19 history	-	-
anti-SARS-CoV-2 N antibodies presence	0	0
COVID-19 vaccinated individuals	36 (100%)	45 (100%)
Type of vaccines	vector—ChAdOx1 (Astra Zeneca)	mRNA—BNT162b2 (Pfizer/BioNTech)

CDs—chronic diseases; a, b—values with different superscripts within the row are significantly different.

## Data Availability

The data presented in this study are available within this article.

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
