# Peer review of "Differences in BNT126b2 and ChAdOx1 Homologous Vaccination Antibody Response among Teachers in Poznan, Poland"

_vaccines, 2023, doi:10.3390/vaccines11010118_

Round 1
Reviewer 1 Report
Estimated Authors of the paper "Differences in BNT126b2 and AZD1222 homologous vaccination antibody response among teachers in Poznan, Poland"
I've read your article with great interest. In this study, Lorent et al have reported on the efficacy of two vaccine formulates (i.e. one mRNA based and one based on adenovirus carrier). The study is affected by some significant shortcomings, and more precisely:
1) the results of this study were largely anticipated: as education workers are not characterized by biological specificities that could have impaired the immune system activation elicited by the vaccines, the assessed vaccines worked exactly how they were expected to. Therefore, the potential contribution of this study to our understanding of the COVID-19 pandemic may be questioned from its roots.
2) the sample is very small: less than 100 cases when over than billions of vaccine doses have been delivered since the beginning of the pandemic can hardly contribute to our understanding of efficacy or side effects of SARS-CoV-2 vaccines
Please understand that I have truly appreciated the data reporting and the approach to the data analysis you relied on, but the paucity of the sample and the lack of significance from a Public Health point of view represents main shortcomings that could be solved only through an extensive assessment of this paper. For example, it would be of some interest to report whether sampled participants have developed or not SARS-CoV-2 infection in the months following the completion of the vaccination schedule, focusing on the severity of symptoms complained.
Anyway, further improvements would be recommended, as follows:
- please include the overall number of education workers who were initially offered to be vaccinated in order to calculate some dropout rate
- please include some information about the vaccination rates in the areas where the study was performed
- please be consistent in the definition of vaccines: in main title, you address Astra Zeneca Vaccine as AZD1222, that is then addressed across the main text as ChAdOx1;
- table 1 should implement a simple univariate comparison in order to appreciate whether the 2 subgroups could be assumed as similar or not
Author Response
Q1 - Estimated Authors of the paper "Differences in BNT126b2 and AZD1222 homologous vaccination antibody response among teachers in Poznan, Poland" I've read your article with great interest. In this study, Lorent et al have reported on the efficacy of two vaccine formulates (i.e. one mRNA based and one based on adenovirus carrier). The study is affected by some significant shortcomings, and more precisely: 1) the results of this study were largely anticipated: as education workers are not characterized by biological specificities that could have impaired the immune system activation elicited by the vaccines, the assessed vaccines worked exactly how they were expected to. Therefore, the potential contribution of this study to our understanding of the COVID-19 pandemic may be questioned from its roots. 2) the sample is very small: less than 100 cases when over than billions of vaccine doses have been delivered since the beginning of the pandemic can hardly contribute to our understanding of efficacy or side effects of SARS-CoV-2 vaccines Please understand that I have truly appreciated the data reporting and the approach to the data analysis you relied on, but the paucity of the sample and the lack of significance from a Public Health point of view represents main shortcomings that could be solved only through an extensive assessment of this paper. For example, it would be of some interest to report whether sampled participants have developed or not SARS-CoV-2 infection in the months following the completion of the vaccination schedule, focusing on the severity of symptoms complained.
A1 – We are grateful for your comments! We agree with the Reviewer, that the vaccinated teachers should respond in the same manner as other occupational groups, described in many other available publications. However, the teachers are in the highest SARS-CoV-2 infection risk group, due to the contact with nonvaccinated children, which as we mentioned, may be a vector for SARS-CoV-2. Therefore, it is worth to analyse the immune response and the vaccine efficiencies against COVID-19 in this group. We added some additional analysis, i.e., the anti-SARS-CoV-2 nucleoprotein antibodies presence, which may describe the vaccine efficiency and thus enhance the value of our findings. We are also aware on the limitations of our study, i.e., relatively small groups, as described in the Conclusions. We also tried to contact the study participants as suggested by the Reviewer, but only a few answered on our additional questions. But still we believe that our findings may be of interests for the readers of Vaccine, especially for the readers of Vaccine Special Issue ‘Vaccination and Its Role in the Prevention of COVID-19 Infection Risk in Workers’
Q2 – Please include the overall number of education workers who were initially offered to be vaccinated in order to calculate some dropout rate
A2 – Thank you for your comment. We have invited approximately 100 teachers from primary schools and 100 academic teachers from Poznan University of Medical Sciences. The missing information was incorporated into the manuscript.
Q3 – Please include some information about the vaccination rates in the areas where the study was performed
A3 – Thank you for your valuable comment! We have added information about vaccination rate in Poznan at the beginning of our study. ‘According to the Polish Ministry of Education and Science, more than 90% of primary and secondary school teachers and 89% of lecturers were administered first dose of the vaccine against COVID-19 in February 2022, while the general vaccination rate in Poznan at the beginning of our study was estimated as 3.84%’.
Q4 – Please be consistent in the definition of vaccines: in main title, you address Astra Zeneca Vaccine as AZD1222, that is then addressed across the main text as ChAdOx1.
A4 – We would like to apologize for being inconsistent with the name of vaccine. We have replaced “AZD1222” with “ChAdOx1” in the main title.
Q5 – Table 1 should implement a simple univariate comparison in order to appreciate whether the 2 subgroups could be assumed as similar or not
A5 – We have compared two analysed groups and found the statistically significant differences only within the numbers of 51-60 y.o. participants and numbers of participants without any comorbidities and chronic diseases. This information was added to the Table 1. The implications of the above-mentioned differences between analysed groups have been already discussed: ‘In our research, in contrast to the BNT162b2 group, systemic reactogenicity was more common and severe after two doses of ChAdOx1. The majority (77%) of ChAdOx1 recipients experienced multiple adverse reactions, such as fever, muscle and joint pain, headache, fatigue, chills, pain at the injection site, diarrhoea or vomiting. (…) Teachers vaccinated with ChAdOx1 are older and many suffer from underlying diseases, so they may pay greater attention to their health after vaccination.’
Reviewer 2 Report
This manuscript, written by Dr. Lorent, an original report, analyzed the effect of vaccination against SARS-CoV-2 in a group of teachers from Poland. The AZ and Pfizer vaccines were tested. The research found that Pfizer provided higher antibody titers and had fewer adverse effects. With time, the antibody counts decreased in both groups, but less in AZ. The manuscript has the limitation of the sample size, and possibly the lack of a control group. Nevertheless, relevant information is provided and it may be worth sharing with the rest of the scientific community. To improve the manuscript, a series of comments were made:
(1) In the abstract, it is written that “Children are among the best vectors to spread respiratory viruses, including emerging variants of SARS-CoV-2, due to the asymptomatic or relatively mild course of infection and simultaneously high titers of pathogens in the respiratory tract.” Could you please confirm that this statement is backed up with scientific evidence? Could you please add a reference?
“Therefore, individuals who have constant contact with children, e.g., teachers, should be vaccinated against COVID-19 as essential workers within the first phases of vaccination campaign. “. Could you please confirm that this statement has been proven? Could you please add a reference?
(2) In the introduction. Could you please provide details of the design of the ChAdOx1 and BNT162b2 design?
For example:
“The AstraZeneca COVID-19 Vaccine is a replication-deficient (i.e. inactivated) chimpanzee adenovirus vector - specifically the ChAdOx1 vector - encoding a trimeric pre-fusion form of the SARS-CoV-2 spike (S) protein.5 Following intramuscular administration these spike proteins are expressed locally, allowing the immune system to mount a neutralizing antibody/cellular immune response. This initial exposure and priming of the immune system subsequently provides protection against future infection”
Sources:
(a) https://go.drugbank.com/drugs/DB15656
(b) https://gsrs.ncats.nih.gov/ginas/app/beta/substances/B5S3K2V0G8
“The Pfizer-BioNTech COVID-19 vaccine (also known as BNT162b2, Tozinameran, and Comirnaty), is one of four advanced mRNA-based vaccines developed through "Project Lightspeed," a joint program between Pfizer and BioNTech.2,3,13 Comirnaty is a nucleoside modified mRNA (modRNA) vaccine encoding an optimized full-length version of the severe acute respiratory syndrome coronavirus 2 (SARS-CoV-2) spike (S) protein. It is designed to induce immunity against SARS-CoV-2, the virus responsible for causing COVID-19.2 The modRNA is formulated in lipid nanoparticles for administration via intramuscular injection in two doses, three weeks apart.1,3
Comirnaty contains nucleoside modified mRNA (modRNA) encapsulated in lipid nanoparticles that deliver the modRNA into host cells. The lipid nanoparticle formulation facilitates the delivery of the RNA into human cells.12 Once inside these cells, the modRNA is translated by host machinery to produce a modified SARS-CoV-2 spike (S) protein antigen, which is subsequently recognized by the host immune system. Comirnaty has been shown to elicit both neutralizing antibody and cellular immune responses to the S protein, which helps protect against subsequent SARS-CoV-2 infection.7,8”
Sources:
(a) https://go.drugbank.com/drugs/DB15696
(b) https://gsrs.ncats.nih.gov/ginas/app/beta/substances/6fefa717-6a4c-435f-a692-46189283764f
(3) Could you please add a figure of the schematic structure of the S protein? A very nice figure is found in the following manuscript: https://doi.org/10.1038/s41401-020-0485-4
(4) Line 82. Could you please add the catalog number of the ELISA kit? I think it is the # TK040 (https://vitrotest.ua/-covid-19-146-.html).
(5) Regarding Table 1. Since there is enough space. These disease abbreviations could be avoided.
(6) Table 1 states that none of the participants had a history of COVID. Could you please explain how this was evaluated?
(7) In Figure 1, it is shown that the ELISA kit measured the levels of IgG antibodies again the S spike protein? Question 1: Do the 2 types of vaccine create in the host the same type subtype of S protein (same nucleotides)? Question 2: Does the ELISA kit identify all subtypes of S spike proteins? Question 3: Is the S protein inmutable?
(8) Regarding Figure 2. I am not sure about the column of “any adverse effects.” The AZ vaccine had more adverse effects than Pfizer. But in this column look opposite.
(9) Regarding section 3.4. After 3 months of vaccination. The IgG levels decreased significantly. Could you please enlarge the y axis of plot 3A to be able to see with more detail the changes in the group of AZ vaccine? Do the levels after 3 months were protective against coronavirus?
(10) This study lacks a control group that did not get vaccinated. Should this be mentioned in the limitations of the conclusions?
Author Response
Q1 – This manuscript, written by Dr. Lorent, an original report, analyzed the effect of vaccination against SARS-CoV-2 in a group of teachers from Poland. The AZ and Pfizer vaccines were tested. The research found that Pfizer provided higher antibody titers and had fewer adverse effects. With time, the antibody counts decreased in both groups, but less in AZ. The manuscript has the limitation of the sample size, and possibly the lack of a control group. Nevertheless, relevant information is provided and it may be worth sharing with the rest of the scientific community.
A1 – We are grateful for you kind opinion!
Q2 - To improve the manuscript, a series of comments were made: In the abstract, it is written that “Children are among the best vectors to spread respiratory viruses, including emerging variants of SARS-CoV-2, due to the asymptomatic or relatively mild course of infection and simultaneously high titers of pathogens in the respiratory tract.” Could you please confirm that this statement is backed up with scientific evidence? Could you please add a reference?
“Therefore, individuals who have constant contact with children, e.g., teachers, should be vaccinated against COVID-19 as essential workers within the first phases of vaccination campaign. “. Could you please confirm that this statement has been proven? Could you please add a reference?
A1 – We have added references according to your suggestion. ‘Children are among the best vectors to spread respiratory viruses, including emerging variants of SARS-CoV-2, due to the asymptomatic or relatively mild course of infection and simultaneously high titres of pathogens in the respiratory tract [1]. Therefore, individuals who have constant contact with children, e.g. teachers, should be vaccinated against COVID-19 as essential workers within first phases of vaccination campaign [2,3].’
- Chou, J.; Thomas, P.G.; Randolph, A.G. Immunology of SARS-CoV-2 Infection in Children. Nat. Immunol. 2022, 23, 177–185, doi:10.1038/s41590-021-01123-9.
- Lorent, D.; Nowak, R.; Roxo, C.; Lenartowicz, E.; Makarewicz, A.; Zaremba, B.; Nowak, S.; Kuszel, L.; Stefaniak, J.; Kierzek, R.; et al. Prevalence of Anti-SARS-CoV-2 Antibodies in Poznań, Poland, after the First Wave of the COVID-19 Pandemic. Vaccines 2021, 9, 541, doi:10.3390/vaccines9060541.
- Monitoring Teacher Vaccination against COVID-19 | UNESCO Available online: https://www.unesco.org/en/articles/monitoring-teacher-vaccination-against-covid-19 (accessed on 15 November 2022).
Q2 – In the introduction. Could you please provide details of the design of the ChAdOx1 and BNT162b2 design? For example: “The AstraZeneca COVID-19 Vaccine is a replication-deficient (i.e. inactivated) chimpanzee adenovirus vector - specifically the ChAdOx1 vector - encoding a trimeric pre-fusion form of the SARS-CoV-2 spike (S) protein.5 Following intramuscular administration these spike proteins are expressed locally, allowing the immune system to mount a neutralizing antibody/cellular immune response. This initial exposure and priming of the immune system subsequently provides protection against future infection”
Sources:
(a) https://go.drugbank.com/drugs/DB15656
(b) https://gsrs.ncats.nih.gov/ginas/app/beta/substances/B5S3K2V0G8
“The Pfizer-BioNTech COVID-19 vaccine (also known as BNT162b2, Tozinameran, and Comirnaty), is one of four advanced mRNA-based vaccines developed through "Project Lightspeed," a joint program between Pfizer and BioNTech.2,3,13 Comirnaty is a nucleoside modified mRNA (modRNA) vaccine encoding an optimized full-length version of the severe acute respiratory syndrome coronavirus 2 (SARS-CoV-2) spike (S) protein. It is designed to induce immunity against SARS-CoV-2, the virus responsible for causing COVID-19.2 The modRNA is formulated in lipid nanoparticles for administration via intramuscular injection in two doses, three weeks apart.1,3
Comirnaty contains nucleoside modified mRNA (modRNA) encapsulated in lipid nanoparticles that deliver the modRNA into host cells. The lipid nanoparticle formulation facilitates the delivery of the RNA into human cells.12 Once inside these cells, the modRNA is translated by host machinery to produce a modified SARS-CoV-2 spike (S) protein antigen, which is subsequently recognized by the host immune system. Comirnaty has been shown to elicit both neutralizing antibody and cellular immune responses to the S protein, which helps protect against subsequent SARS-CoV-2 infection.7,8”
Sources:
(a) https://go.drugbank.com/drugs/DB15696
(b) https://gsrs.ncats.nih.gov/ginas/app/beta/substances/6fefa717-6a4c-435f-a692-46189283764f
A2 – We are grateful for your valuable comments. We added short introduction on the different types of vaccines in the Introduction accordingly. ‘Both BNT162b2 and ChAdOx1 elicit immune response mainly against the receptor-binding domain (RBD) of spike (S) protein. However, the S protein consists additionally following domains: N-terminal domain (NTD), fusion peptide (FP), two heptad repeats (HR1 and HR2), transmembrane domain (TM) and cytoplasmic tail (CT), which may be also a target for antibodies produced as a result of vaccination. The structure of the SARS-CoV-2 S protein is well presented by Huang et al. [13]. It should be also highlighted that different vaccine types utilize different nucleotide sequences as well as delivery methods. ChAdOx1 encodes a full-length S protein identical to SARS-CoV-2 wild-type (wt) strain, i.e., Wuhan-Hu-1 [14]. In contrast, the BNT126b2 contains nucleoside modified mRNA with pseudouridines, and differs in two amino acids from the wt S protein sequence, to stabilize the construct in the cellular environment and to stabilize the prefusion conformation of S protein, respectively [15]. The ChAdOx1 is delivered in the replication-deficient chimpanzee adenovirus-vector, whereas BNT162b2 contains nucleoside modified mRNA sequence encapsulated in lipid nanoparticles [14,15]. Both vaccines are administrated intramuscularly and have been shown to be effective in delivery of viral antigens, thereby stimulating host immune system to recognize SARS-CoV-2.’
Q3 – Could you please add a figure of the schematic structure of the S protein? A very nice figure is found in the following manuscript: https://doi.org/10.1038/s41401-020-0485-4
A3 – We would like to thank for the comment! Due to the perfect presentation of S protein structure in the above-mentioned manuscript, we decided to cite the paper instead of adding additional Figure.
Q4 – Line 82. Could you please add the catalog number of the ELISA kit? I think it is the # TK040 (https://vitrotest.ua/-covid-19-146-.html).
A4 – The information has been added as suggested. ‘Anti-SARS-CoV-2 IgG antibody levels after vaccination were measured using quantitative SARS-CoV-2 IgG QuantiSpike ELISA assay (TK040, Vitrotest Europe, Wroclaw, Poland) targeting spike (S) protein.’
Q5 – Regarding Table 1. Since there is enough space. These disease abbreviations could be avoided.
A5 – We are grateful for your comment and changed the abbreviations into full names of comorbidities and chronic diseases.
Q6 – Table 1 states that none of the participants had a history of COVID. Could you please explain how this was evaluated?
A6 – The history of COVID-19 was evaluated on the basis of study participants self-reports including the occurrence of flu-like symptoms in the last 6 months and the results of the SARS-CoV-2 diagnostic tests. Additionally, we determined the presence of the anti-SARS-CoV-2 nucleocapsid (N) protein antibodies with rapid cassette tests, i.e., 2019-novel coronavirus IgG/IgM antibody detection kit (Vazyme). The anti-SARS-CoV-2 N antibodies are produced only after coronavirus infection. We did not find any anti-SARS-CoV-2 N antibodies among study participants at two weeks, and at three months after vaccination. The description on the evaluation of COVID-19 history was added to the manuscript.
‘To test whether the study participants were infected with SARS-CoV-2 we used rapid cassette tests: the 2019-novel coronavirus IgG/IgM antibody detection kit (Vazyme), which detects anti-SARS-CoV-2 N antibodies generated only after natural infection.’
Q7 – In Figure 1, it is shown that the ELISA kit measured the levels of IgG antibodies again the S spike protein? Question 1: Do the 2 types of vaccine create in the host the same type subtype of S protein (same nucleotides)? Question 2: Does the ELISA kit identify all subtypes of S spike proteins? Question 3: Is the S protein inmutable?
A7 – We are grateful for your comment. As we mentioned in the description of analyzed vaccines in the Introduction section, the BNT126b2 sequence differs from the ChAdOx-1 in two amino acids: ‘ChAdOx1 encodes a full-length S protein identical to SARS-CoV-2 wild-type (wt) strain, i.e., Wuhan-Hu-1 [14]. In contrast, the BNT126b2 contains nucleoside modified mRNA with pseudouridines, and differs in two amino acids from the wt S protein sequence, to stabilize the construct in the cellular environment and to stabilize the prefusion conformation of S protein, respectively [15].’ Therefore, we assume that the antibodies produced after vaccination are similar and there are no significant differences. In addition, the chosen ELISA kit, produced by Vitrotest Europe, uses the whole wt S protein and thus, should identify all antibodies generated after vaccination. AS it was stated, both analyzed vaccine contains the wt S sequence, used also in the chosen ELISA kit. On the market, there are ELISA kits which are based only on the RBD of SARS-CoV-2 S protein, which may omit the developed antibodies targeting other parts of S protein, and thus were not chosen for our study. However, it should be noted, that the SARS-CoV-2 mutates rapidly, and new genetic variants emerged, i.e. SARS-CoV-2 omicron variant. Therefore, the new vaccines against omicron variant were developed by Pfizer and Moderna, and currently are recommended as a second booster dose. The anti-SARS-CoV-2 antibodies panel after second booster vaccine dose may differ from the antibodies panel developed after first two doses.
Q8 – Regarding Figure 2. I am not sure about the column of “any adverse effects.” The AZ vaccine had more adverse effects than Pfizer. But in this column look opposite.
A8 – We would like to apologize for being unclear. The Figure 2 was changed accordingly.
Q9 – Regarding section 3.4. After 3 months of vaccination. The IgG levels decreased significantly. Could you please enlarge the y axis of plot 3A to be able to see with more detail the changes in the group of AZ vaccine? Do the levels after 3 months were protective against coronavirus?
A9 – Thank you for your suggestion. The Figure 3 was modified accordingly. Regarding the antibodies titers protective against coronavirus, according to our knowledge so far no one did not estimate the range of antibodies level, which protects against COVID-19. Therefore, as we mentioned in the Discussion, the booster vaccine doses are recommended. ‘Due to the above-described depletion of anti-SARS-CoV-2 S antibodies and thus potential weaker protection from severe COVID-19, as well as the emergence of the SARS-CoV-2 omicron variant, a third vaccine dose, the so-called ‘booster’, is recommended by EMA, the European Center for Disease Prevention and many other scientific and medical societies’. In addition, as we mentioned in the manuscript, the analysis of anti-SARS-CoV-2 antibodies level is the easiest way to estimate the immune reaction, but it must be highlighted that there are additional strategies to protect from the diseases: ‘Studying other mechanisms such as memory T cells may enhance our understanding of vaccine-induced immunity’.
Q10 –This study lacks a control group that did not get vaccinated. Should this be mentioned in the limitations of the conclusions?
A10 – We are grateful for your valuable comment. Unfortunately, as we mentioned in the manuscript ‘more than 90% of primary and secondary school teachers and 89% of lecturers were fully vaccinated against COVID-19’. This caused that it was impossible to find the non-vaccinated individuals and to convince them to participate in our study. However, we fully agreed with the Reviewer and add this point to the limitations in the Discussion section. ‘Lastly, due to the very high vaccination rate, we were unable to collect samples from the non-vaccinated teachers, which could serve as a control group in comparison of the vaccine effectiveness and SARS-CoV-2 infection risk.’
Round 2
Reviewer 1 Report
I've appreciated the considerable efforts paid by study Authors.
As a consequence, I'm endorsing the eventual acceptance of this study.